# Monitoring of Animal Feed Contamination by Mycotoxins: Results of Five Years of Official Control by an Accredited Italian Laboratory

**DOI:** 10.3390/microorganisms12010173

**Published:** 2024-01-15

**Authors:** Cinzia Franchino, Valeria Vita, Marco Iammarino, Rita De Pace

**Affiliations:** Istituto Zooprofilattico Sperimentale di Puglia e Basilicata, Via Manfredonia 20, 71121 Foggia, Italy; cinzia.franchino@izspb.it (C.F.); valeria.vita@izspb.it (V.V.)

**Keywords:** Aflatoxin, Deoxynivalenol, ELISA, Fumonisins, HT_2_, mycotoxins, Ochratoxin, T_2_

## Abstract

Mycotoxin contamination of animal feed is a complex issue in both animal wellness and food safety. The most diffused mycotoxins subject to the official control of animal feed are Aflatoxin B1 (AF), Zearalenone (ZEA), Deoxynivalenol (DON), Ochratoxin A (OCRA), Fumonisins (FUMO), and T-2/HT-2 toxins. This work describes the results of five years of monitoring focused on the evaluation of mycotoxin contamination of animal feed. Analytical determinations were carried out by means of accredited ELISA. The obtained results showed a non-alarming scenario, with several samples resulting as “non-compliant” according to the Maximum Residue Limits (MRLs) set in European Regulation No. 574/2011. Out of 722 analyzed samples coming from 2 Italian regions, Apulia and Basilicata, 14 samples were characterized by mycotoxin concentrations higher than related MRL; in particular, 5, 4, and 5 non-compliant samples for DON, AF, and ZEA, respectively. This study also evaluated the possible correlations between mycotoxin type and feed use with a special focus on animal sensitivity to mycotoxins.

## 1. Introduction

The possible presence of different kinds of mycotoxins in vegetable matrices is a worldwide concern with several negative impacts on animal wellness and the safety of animal-source products.

The mycotoxins considered the most significant in food safety monitoring programs are Aflatoxins (AFs), which are secondary metabolites mainly produced by certain fungi such as *Aspergillus flavus*, *Aspergillus parasiticus*, and *Aspergillus nomius*; Deoxynivalenol (DON), Zearalenone (ZEA), and Fumonisins (FUMO) produced by *Fusarium* fungi; and Ochratoxin A (OCRA) and T-2/HT-2 toxins produced by *Penicillium* and *Aspergillus* fungi. The main toxins produced by these fungi are thermostable, so they can exist as residue in animal feed after production for long durations [1,2,3,4].

In developed countries, acute mycotoxicosis in animal husbandry has become quite rare as a result of improved animal feed hygiene. Unfortunately, it is not possible to state the same for subclinical mycotoxicosis due to the presence of low concentrations of mycotoxins in food and feed, which are sometimes capable of seriously affecting different economic aspects. In particular, the immunosuppressive effects of different mycotoxins can also expose animals to pathologies from not particularly aggressive germs or make the response to vaccines less effective. Regarding meat production, in some cases, these minor toxicological consequences, while not preventing the animal from reaching slaughter, can significantly compromise the quality of final products.

AFs were defined as potent carcinogenic compounds by IARC in 2012 [5]. Moreover, they may be responsible for several types of damage to different organs, causing hepatotoxicity, immunosuppression, coagulation alteration, etc. [6,7]. Regarding AF toxicity and animal health, young poultry, rabbits, and swine are more susceptible. In these species, AFs can cause anorexia, hemorrhages, edema, and jaundice.

DON is a mycotoxin mainly produced by *Fusarium graminearum* [8]. It is involved in protein synthesis inhibition, with high toxicity on the intestinal apparatus and the ability to cause immune deficiency. Several studies reported that poultry, swine, and rabbits are the most susceptible species to this mycotoxin and that this kind of poisoning is appreciable since animals suffer from vomiting and refuse food [9].

ZEA is a toxin primarily produced by *Fusarium graminearum*, and in lower quantities by other *Fusarium* species such as *culmorum*, *verticillioides* (moniliforme), *sporotrichioides*, *semitectum*, *equiseti*, and *oxysporum* [10]. This toxin is responsible for estrogenic syndrome, reproductive toxicity, and immunotoxicity, especially in swine and rabbits, with consequent hyperestrogenism, prolapses, and abortions [11].

FUMO is a group of toxins (mainly four compounds) produced by *Fusarium verticilliodes* and *Fusarium proliferatum*, which can cause nephrotoxic and hepatotoxic effects [12]. Equine and swine are the most susceptible species to this toxin, and jaundice, edema, and severe dyspnea represent the main toxic effects.

OCRA is a toxin that was first isolated as a secondary metabolite from *Aspergillus ochraceus* in 1965 [13]. Other than *Aspergillus*, *Penicillium* can also produce this toxin [14], which is characterized by nephrotoxic, immunotoxic, and genotoxic effects, especially in chicken and swine. Polydipsia, polyuria, colic, edema, and diarrhea are the most evident toxic effects [15].

T-2 and HT-2 toxins belong to Trichothecenes, a family of mycotoxins (more than 200 compounds) produced by many species of fungi, such as *Fusarium*, *Trichothecium*, *Trichoderma*, *Myrothecium*, *Cephalosporium*, *Spicellum*, and *Stachybotrys* [16]. These toxins can have a negative impact on heart muscles, nerves, and the immune system [17]. As per DON, poultry, swine, and rabbits represent the most susceptible species to this toxin, and oral ulcers and melena are the main toxic effects.

All mycotoxins listed above are characterized by different absorptions, metabolisms, and kinetics. Consequently, all legislative aspects related to mycotoxin contamination in animal feed have taken into account several parameters, with specific limits for each mycotoxin as linked to animal species, age, zootechnical function, etc. [18,19,20] (Table 1).

Several studies have demonstrated a positive correlation between climate change and the increase in contamination by molds and related metabolites in different matrices [21]. The main factors that seem to influence such an increase are temperature and CO_2_ increases and drought stress [22]. Consequently, the official control of animal feed contamination by mycotoxins has gained importance in recent years. As an example, studies conducted on animal feed samples commercialized in European-Mediterranean and Asia/Pacific areas demonstrated that more than half of the samples were contaminated by mycotoxins at levels higher than the respective legal limits [21,23].

In this paper, the results obtained from 5 years of official control activity carried out at an Italian accredited laboratory focused on the analytical determination of AF, ZEA, DON, OCRA, FUMO, and T-2/HT-2 mycotoxins in 722 samples of animal feed are described. The specific correlations between mycotoxin and animal species were also investigated.

## 2. Materials and Methods

All samples described in this study were supplied by local veterinary services within the official control activity of animal feed, developed during the last years (2018–2022), following the provision described in European Regulation No. 152/2009/EC [24]. These instructions are conceived taking into account the fact that mycotoxin contamination is often developed patchily on the whole feed batch. Thus, in order to obtain reliable findings, the homogeneity of the collected sample has to be assured considering different withdrawal points and large amounts of the matrix to be homogenized properly in the second step using suitable machines able to obtain the right granulometry.

The analytical determinations were carried out by following the instructions supplied by kit producers. The kit for the determination of AF was supplied by Tecna (Gold Standard Diagnostics s.r.l., Trieste, Italy), for FUMO determination by Euroclone (Euroclone S.p.A., Pero, Milan, Italy), and for OCRA, DON, ZEA, and T-2/HT-2 by R-Biopharm (R-Biopharm Italia s.r.l., Melegnano, Milan, Italy). The capability of ELISA to carry out a large number of determinations at a low cost, rapidly, and with high sensitivity makes the choice of such a technique extremely affordable for laboratories in charge of food inspection, especially if compared to other approaches such as chromatographic techniques, which are better suited for confirmatory analysis purposes [2].

All these procedures were fully validated and accredited by the Italian Organism for laboratory accreditation ACCREDIA, according to the ISO/IEC/17025 [25]. The following validation parameters were evaluated, in accordance with the Regulation (EU) 2021/808 [26]: limit of detection (LOD), selectivity, CC_β_, accuracy (as recovery percentage and precision), and robustness.

The LOD is “the lowest amount of analyte in a sample that can be detected with (stated) probability, although perhaps not quantified as an exact value”, while the limit of quantification (LOQ) is “the lowest amount of analyte in a sample that can be quantitatively determined with stated acceptable precision and trueness, under stated experimental conditions” [27]. As they are “screening” techniques not usually recognized for quantification purposes during the official control of food and feed, only LOD is evaluated during the validation procedure. The LODs of different procedures adopted to detect mycotoxins are those indicated by kit manufacturers. Based on these “cut-off” values, method selectivity and CC_β_ “Detection Capability” were evaluated. CC_β_ is “the smallest content of the analyte that may be detected, identified and/or quantified in a sample with an error probability of β”. The β error, which should be <5%, is “the probability that the sample is truly non-compliant even though a compliant measurement has been obtained”. There are different procedures for the determination of CC_β_ depending on the screening target concentration (SCT). For instance, if the SCT is set at half the maximum residue limit (MRL) or lower, the demonstration that CC_β_ is less than the MRL is obtained by analyzing twenty samples and then verifying the occurrence of one or no false-compliant results [27]. The CC_β_ of methods used in this study was ascertained by analyzing 20 not contaminated feed samples and the same spiked at the level of interest (corresponding to the LOD).

Method accuracy is “the closeness of agreement between a test result and the accepted reference value” and “it is determined by determining trueness and precision”. The trueness represents the consistency between the mean value obtained from a large series of experiments and a value adopted as reference (true value). Usually, it is expressed as the percentage of variance between the measured value and the reference one. When certified materials are available, it is correct to talk in terms of trueness. Otherwise, it is necessary to proceed with the estimation of the recovery percentage by analyzing spiked samples. The precision represents the degree of agreement among the results of independent tests, obtained under defined conditions. It is expressed in terms of inaccuracy as the standard deviation of the measurements. Usually, it is expressed as the percentage of variance between the measured value and the reference one.

Method precision was evaluated by analyzing “blank” feed samples” spiked at three increasing levels of fortification, corresponding to 1/2, 1, and 3/2 the legal limit, under repeatability conditions. The coefficient of variation (CV%) and the mean recovery percentage were calculated to verify precision and trueness, respectively, and these values were evaluated for their compliance with Internationally recognized standards, such as the Horwitz equation [26].

Regarding method robustness, which is “a measure of its capacity to remain unaffected by small, but deliberate variations in method parameters and provides an indication of its reliability during normal usage”, this parameter was studied under “minor changes” conditions, which encompasses analyzing samples fortified at a concentration corresponding to the legal limit and making some slight changes to the sample preparation procedure. These changes can be different centrifugation times and temperatures, times of extraction, temperatures and storage times of the final extract before analysis, etc. The Youden approach was adopted for these trials [26].

All parameters were verified for their homoscedasticity at different fortification levels by means of one-way ANOVA, Student’s *t*-test, and Fisher test (*p* < 0.05).

The most important validation parameters that characterize these methods are presented in Table 2. Regarding the method accuracy, mean recovery percentages, and precision (as CV%), these resulted in the range of 81–122% and 5.5–9.5%, respectively, assuring the results’ reliability. The analytical performances of the applied methods are routinely verified by means of control charts and proficiency test rounds, evaluated as z-scores (lower than 2 and equal to 1.7, 0.1, 0.3, 0.3, 0.8, and 0.1 for AFLA, OCRA, DON, ZEA, FUMO, and T-2/HT-2, respectively, during the last round) [27]. Left-censored data were managed following the “middle-bound” approach, which is considered a protective measure in food safety and environment sustainability, as described in the “Rapporti ISTISAN 04/15” document published by the Italian Institute of Health [28]. Thus, all values lower than the respective limit of detection (LOD) during data elaboration were counted as a concentration equal to LOD/2.

Overall, 722 samples of animal feed were collected and analyzed. The first subdivision of samples was made as follows: raw materials (226 samples), compound/complex feeds (388 samples), and complementary feeds (108 samples). In Table 3, an accurate description of such samples, subdivided by mycotoxin and animal species, is reported.

From the total weight of each sample, when properly homogenized, in the range of 250–1000 g, 5 g samples were weighed for the determination of AF, ZEA, DON, FUMO, and T-2/HT-2, while 2 g was weighed for OCRA analysis by means of ELISA. Each analysis was repeated twice, and the final concentration of mycotoxin was calculated as the mean of the two measurements.

It is worth highlighting that all feed samples with a mycotoxin concentration higher than the respective legal limit were submitted to confirmatory analysis by means of validated and accredited chromatography/mass spectrometry techniques. More specifically, 2 Italian laboratories served as “confirmatory” units. HPLC/FLD was used to confirm N.C. samples for AFLA and ZEA, while UPLC/MS was used to confirm DON non-compliance. All concentrations were confirmed with a quantification error of <10%. As they were supplied by other laboratories, these data are not shown.

## 3. Results

### Animal Feed Contamination by Mycotoxins

All results obtained by analyzing 722 samples of animal feed, subdivided by mycotoxin and use, are reported in Table 3.

Looking at the percentages of “contaminated” samples, which are samples with mycotoxin percentages higher than the respective LOD, the highest probability was verified for T-2/HT-2 toxins with percentages higher than 70% and equal to 100% in samples of feed for equine, rabbits, and swine.

The percentages related to ZEA, OCRA, and FUMO contaminations were lower (<42.5%) and comparable, while a slightly higher level of contamination was registered for AF (in the range of 35% (poultry)–58% (cow)). The lowest percentage of contaminated samples was verified for DON (<22%).

Regarding “non-compliant” (N.C.) samples, 14 samples overall (out of 722 analyzed, equal to 1.9%) showed mycotoxin concentrations higher than the related legal limits. The highest number of N.C. samples was contaminated by levels of DON and ZEA higher than the respective legal limits. More specifically, N.C. samples for DON consisted of one and four samples of feed for cow and swine, respectively, with concentrations equal to 6.87 mg kg^−1^ in the cow’s feed sample and in the range of 0.9–5.0 mg kg^−1^ in the four samples of feed for swine (Table 3). Apart from the sample with 1.3 mg kg^−1^ of DON, composed of corn, the other samples were complementary feeds with different compositions. Corn and barley were the main ingredients in samples with DON concentrations of 6.87 and 1.6 mg kg^−1^, while corn and wheat were the main components of feeds with DON concentrations equal to 5.0 and 0.9 mg kg^−1^. Relating to ZEA, the following concentrations higher than legal limits were detected: 0.57, 0.73, and 1.23 mg kg^−1^ (three samples of feed for cow), 0.51 mg kg^−1^ (one sample of feed for sheep/goat), and 0.51 mg kg^−1^ (one sample of feed for swine). Three samples (with ZEA concentrations equal to 0.57, 0.51, and 0.51 mg kg^−1^) were composed of corn, and the other two N.C. samples were complementary feeds where corn and wheat were the main ingredients. The remaining four N.C. samples showed concentrations of AF higher than the legal limit. In particular, there were two samples of feed for cows (12.6 and 15.5 µg kg^−1^), only composed of corn, one sample of complementary feed for sheep/goat (7.16 µg kg^−1^), where corn and wheat were the main ingredients, and one sample of complementary feed for swine (5.71 µg kg^−1^) with barley and corn as the main components. No samples resulted as N.C. relating to T-2/HT-2, OCRA, and FUMO mycotoxins.

The possible co-occurrence of different mycotoxins in the same sample is an emerging and very interesting topic worthy of research. In this study, not all mycotoxins were analyzed for each sample, so this kind of evaluation can only be used to make a general remark. Taking into account the samples analyzed during the last year of monitoring, 1 sample of feed for dairy cows resulted in being contaminated by four mycotoxins (DON-ZEA-OCRA-T-2/HT-2) and 12 samples by three mycotoxins. These samples were subdivided as follows: for DON-FUMO-T-2/HT-2, DON-OCRA-T-2/HT-2, and ZEA-FUMO-OCRA, 2 samples each of feed for dairy cows, corn flour, and feed for cows, respectively; for ZEA-FUMO-T-2/HT-2 and FUMO-OCRA-T-2/HT-2, 6 samples of feed for cows and dairy cows; 28 samples by 2 mycotoxins, where the simultaneous presence of FUMO and T-2/HT-2 was the most frequent combination.

In Figure 1, a graphical elaboration of the obtained data, subdivided by mycotoxin and animal species, is shown.

## 4. Discussion

### 4.1. Contamination by Mycotoxins in Feed for Different Species

Looking at Figure 2, it is possible to appraise how the mean concentration of each mycotoxin (evaluated using the middle-bound approach) resulted close to the respective legal limit. This evaluation is essential, in addition to the previous count of samples percentage > LOD, in order to obtain an overall “view” of feed contamination.

Some mean levels are worthy of attention due to both food safety and animal wellness reasons. Swine feed was characterized by some mycotoxin levels higher than other feed types and close to the legal limits. In particular, the ZEA mean concentration detected in feed for swine (0.179 mg kg^−1^) was higher than the legal limit set for piglets and close to that set for fattening pigs (0.1 and 0.25 mg kg^−1^, see Table 1). Also, the mean concentration of DON detected in feed for swine (0.5 mg kg^−1^) resulted in more than half the legal limit (0.9 mg kg^−1^). Another significant finding is related to T-2/HT-2 mean contamination. Indeed, all mean levels registered during this monitoring, in the range of 107.45–125.26 µg kg^−1^ related to sheep/goat and rabbit/swine feed, respectively (Table 3), resulted in being close or corresponding to half of the legal limit defined in the Commission’s Recommendation of 27 March 2013 (250 µg kg^−1^) [19]. All other mean concentrations of mycotoxins detected in different types of animal feed can be considered low since they were quite far from the related legal limit and therefore did not represent a significant concern in either food safety or animal health.

As stated above, previous studies demonstrated that some particular animals seem more susceptible to specific mycotoxins. In this regard, this monitoring was useful to appraise the specific correlation between animal feed and mycotoxins. Referring again to Figure 2, where the most susceptible species to each mycotoxin are pointed out in red boxes, it is possible to confirm that the contamination levels of feed for swine are the worthiest of attention. Indeed, swine is very susceptible to three mycotoxins, namely ZEA, DON, and T-2/HT-2, for which the mean concentrations detected during this survey resulted in being not so far from the related legal limit. Another important finding from this perspective is related to T-2/HT-2 mean concentrations registered in feed for rabbits and poultry, which are equal to 125.26 and 111.48 µg kg^−1^, respectively (Table 3). Indeed, with these two animal species particularly susceptible to T-2/HT-2 toxins, these high levels so close to the half of legal limit (250 µg kg^−1^) can be considered worthy of attention.

### 4.2. Comparison to the Literature

The topic “mycotoxins contamination of animal feed” was studied in some interesting papers published in recent years. In this regard, a comparison between the findings of this survey and other data available from other countries can be very interesting.

Kosicki et al. [29] analyzed 1384 samples of animal feed in Poland over 4 years (2011–2014). These authors found a comparable percentage of contaminated samples regarding T-2/HT-2, FUMO, and OTA, while the percentage of samples contaminated by DON and ZEA was much higher (95% and 96%, respectively) with respect to this study. Only the AF contamination percentage obtained in this study (~37%) was higher than that reported by Kosicki et al. (5%). The percentage of “non-compliant” samples was also comparable between this study (1.8%) and that of Kocak et al. (1.7%). These results were substantially confirmed by Twarużek et al. [30] in a similar study, carried out in Poland during the years 2015–2020 on 3980 samples of animal feed. In this case, the percentage of “non-compliant” samples was slightly lower (1.2%).

In 2022, Ferrari et al. [31] published results obtained from monitoring Aflatoxin B1 in 10,280 samples of animal feed and forages collected in Northern Italy during the years 2013–2021. The authors found a percentage of contamination (samples > LOD: 1.5 µg kg^−1^) of 14.7%, which is lower than the percentage obtained in this survey (~37%). However, the LOD of this monitoring (1.0 µg kg^−1^) being lower could justify the higher percentage of detection. The percentage of “non-compliant” samples reported by Ferrari et al. was much higher (5.7%) than this study (0.5%). This difference may be due to the type of feed collected since these authors only analyzed raw materials, while in this study, the majority of samples (496 out of 722, equal to 68.2%) were composed of compound/complex and complementary feeds. Indeed, it is well known that feed processing can lead to the decontamination of raw materials [32,33,34,35].

Gruber-Dorninger et al. [36] published a comprehensive paper focused on the evaluation of “non-compliant” samples for mycotoxins analyzed in 100 countries from 2008 to 2017. The percentage of samples with AF concentrations higher than the legal limits obtained during the present study (0.5%) was comparable to that obtained for Northern Europe (0.4%). Regarding ZEA, the percentage of this study (0.7%) corresponded to that of Oceania and very closely to North America (0.6%). If compared to data from southern Europe, the percentage of contamination obtained from this monitoring resulted as comparable for ZEA and OCRA, lower for DON and FUMO, and higher for AFLA and T-2/HT-2.

Looking at the findings obtained from this study, some mycotoxin/animal combinations seem to be of major concern. In this regard, advanced decontamination strategies for reductions in DON, ZEA, and T-2/HT-2 in feed for swine, poultry, and rabbits should be developed and applied. In this regard, effective treatments for removing/detoxifying DON in animal feed based on ion-exchanged zeolites absorption [37], *Rhodotorula glutinis, B. subtilis* ZZ, and other microorganism and enzymatic actions [38,39], and ozone [40] were reported during recent years [41]. With regard to ZEA, interesting approaches to its removal from animal feed based on *L. plantarum* BCC 47723 [42] and conidia of *Aspergillus japonicus* [43] activities were recently published. Finally, an effective method for T-2 toxin removal from animal feed based on the activity of the non-protein material of the extracellular fraction of *Lactococcus lactis* CAMT22361 cells was reported by Zhou et al. [44].

## 5. Conclusions

This work describes the results of five years of monitoring focused on the evaluation of Aflatoxin B1 (AF), Zearalenone (ZEA), Deoxynivalenol (DON), Ochratoxin A (OCRA), Fumonisins (FUMO), and T-2/HT-2 toxins in 722 samples of animal feed. Fourteen samples were characterized by mycotoxin concentrations higher than related MRL. In particular, five, four, and five non-compliant samples were observed for DON, AF, and ZEA, respectively. Corn was the most frequently present raw material in these samples.

This study particularly focused on possible correlations between mycotoxin type and animal susceptibility. In this regard, the findings worthy of attention were related to swine since it is very susceptible to three mycotoxins, namely ZEA, DON, and T-2/HT-2, and the mean concentrations detected during this survey were not far from reaching the related legal limit. Moreover, relating to T-2/HT-2, the mean contamination registered in feed for rabbits and poultry, which was close to half the legal limit, is worth mentioning since these two animal species are particularly susceptible to these toxins.

By virtue of the foregoing, special precautions should be applied during the manufacturing and storage of feed for swine, rabbits, and poultry, in order to obtain an overall decrease in contamination by certain mycotoxins, particularly ZEA, DON, and T-2/HT-2. Finally, regarding official control, tighter control over these activities should be planned for these types of animal feed/mycotoxin combinations.

## Figures and Tables

**Figure 1 microorganisms-12-00173-f001:**
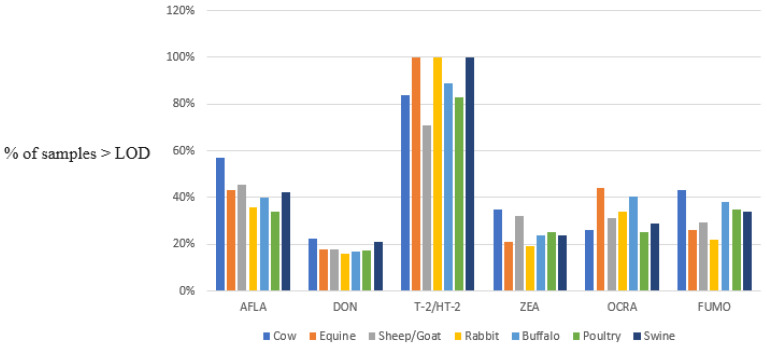
Percentages of samples of animal feed with quantifiable amounts of different mycotoxins.

**Figure 2 microorganisms-12-00173-f002:**
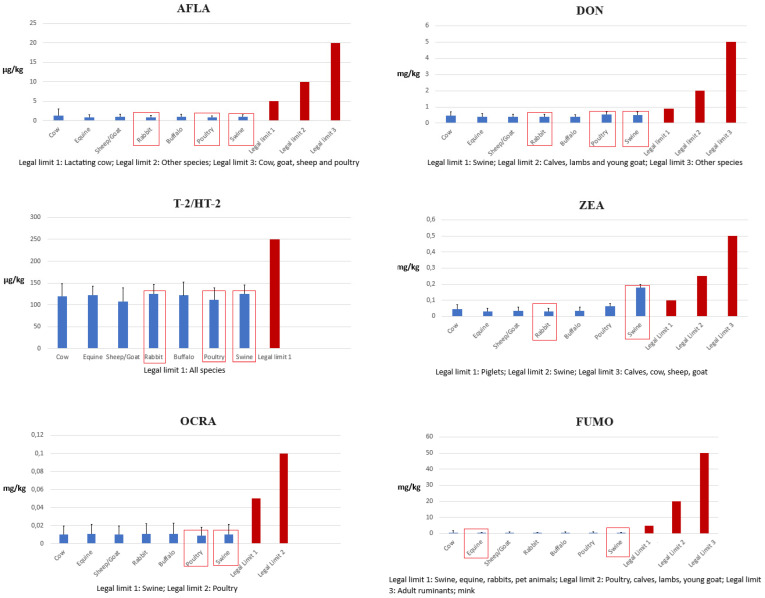
Mean levels of mycotoxins in samples of feed for different animal species (blue); legal limits (red). In the red boxes are the most susceptible animal species to the specific mycotoxin.

**Table 1 microorganisms-12-00173-t001:** Legal limits for major mycotoxins in animal feed as set by the European Commission.

Mycotoxin	Animal Species	Legal Limit (mg kg^−1^)
AF	Cow	0.005
	Other species	0.010
	Cow, goat, sheep and poultry	0.020
DON	Swine	0.9
	Calves, lambs and young goat	2
	Other species	5
ZEA	Piglets	0.1
	Fattening pigs	0.25
	Calves, cow, sheep and goat	0.5
OCRA	Swine	0.05
	Poultry	0.1
FUMO	Swine, equine, rabbits and pet animals	5
	Poultry, calves, lambs and young goat	20
	Adult ruminants and mink	50
T-2/HT-2	All species	0.25

**Table 2 microorganisms-12-00173-t002:** Validation parameters of analytical methods adopted in this study.

Mycotoxin	LOD (µg kg^−1^)	SD *	CV (%) *	Recovery (%) *
Aflatoxin B1 (AF)	1.0	0.304	9.5	98.0%
Deoxynivalenol (DON)	555	0.327	9.0	90.5%
Fumonisins (FUMO)	0.2	0.527	6.5	117.0%
Ochratoxin A (OCRA)	12.5	0.021	5.5	106.5%
T-2/HT-2 toxins	75.0	12.165	6.5	102.0%
Zearalenone (ZEA)	8.0	0.029	8.5	97.0%

LOD = Limit of detection; * SD = Standard Deviation; CV = Coefficient of variation. Evaluated as the mean of three fortification levels (n = 6 replicates for each level).

**Table 3 microorganisms-12-00173-t003:** Results obtained by analyzing 722 samples of animal feed.

		AF (µg kg^−1^)	DON (mg kg^−1^)	T-2/HT-2 (µg kg^−1^)	ZEA (mg kg^−1^)	OCRA (mg kg^−1^)	FUMO (mg kg^−1^)	Total (N° of N.C.)
COW	N° of samples	380	288	55	288	288	288	
	<LOD	164	223	9	188	214	163	380
	Min–Max	0.5–15.5	0.28–6.87	37.5–214.3	0.004–0.730	0.006–0.140	0.1–9.3	(6)
	Mean	1.3	0.46	119.3	0.045	0.010	0.4	
	N° of N.C.	2 (12.6–15.5)	1 (6.87)	0	3 (0.57–0.73–1.23)	0	0	
EQUINE	N° of samples	51	34	7	34	34	34	
	<LOD	29	28	0	27	19	25	51
	Min–Max	0.5–2.2	0.277–1.730	99.96–179.14	0.004–0.510	0.006–0.035	0.10–0.73	(0)
	Mean	0.94	0.411	121.51	0.029	0.011	0.17	
	N° of N.C.	0	0	0	0	0	0	
SHEEP/GOAT	N° of samples	79	71	14	71	71	71	
	<LOD	43	58	4	48	49	50	79
	Min–Max	0.50–7.16	0.277–1.730	37.50–179.14	0.004–0.510	0.006–0.035	0.1–6.3	(2)
	Mean	1.04	0.404	107.45	0.033	0.010	0.4	
	N° of N.C.	1 (7.16)	0	0	1 (0.510)	0	0	
	N° of samples	39	32	5	32	32	32	
RABBIT	<LOD	25	27	0	26	21	25	39
	Min–Max	0.5–2.2	0.277–1.730	99.96–179.14	0.004–0.510	0.006–0.035	0.10–0.73	(0)
	Mean	0.9	0.415	125.26	0.030	0.011	0.17	
	N° of N.C.	0	0	0	0	0	0	
	N° of samples	48	42	9	42	42	42	
BUFFALO	<LOD	29	35	1	32	25	26	48
	Min–Max	0.5–4.9	0.277–1.730	37.50–181.64	0.004–0.510	0.006–0.035	0.10–3.08	(0)
	Mean	1.1	0.407	122.38	0.033	0.011	0.34	
	N° of N.C.	0	0	0	0	0	0	
	N° of samples	47	40	6	40	40	40	
POULTRY	<LOD	31	33	1	30	30	26	47
	Min–Max	0.5–2.2	0.28–4.68	37.50–179.14	0.004–0.730	0.006–0.035	0.10–2.99	(0)
	Mean	0.86	0.52	111.48	0.061	0.009	0.324	
	N° of N.C.	0	0	0	0	0	0	
SWINE	N° of samples	47	38	5	38	38	38	
	<LOD	27	30	0	29	27	25	47
	Min–Max	0.50–5.71	0.3–5.0	99.96–179.14	0.004–0.510	0.006–0.035	0.10–1.75	(6)
	Mean	1.06	0.5	125.26	0.179	0.010	0.26	
	N° of N.C.	1 (5.71)	4 (0.9–1.3–1.6–5.0)	0	1 (0.510)	0	0	
TOTAL		691	545	101	545	545	545	
% samples > LOD		49.6%	20.4%	85.1%	30.3%	29.4%	37.6%	
N° of N.C. (%)		4 (0.6%)	5 (0.9%)	0 (0%)	3 (0.5%)	0 (0%)	0 (0%)	

LOD = Limit of Detection (see Table 2); N.C. = Non-compliant.

## Data Availability

Data are available upon request.

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
