# Peer review of "Monitoring of Animal Feed Contamination by Mycotoxins: Results of Five Years of Official Control by an Accredited Italian Laboratory"

_microorganisms, 2024, doi:10.3390/microorganisms12010173_

Round 1

Reviewer 1 Report

Comments and Suggestions for Authors

Manuscript ID: microorganisms-2780545

Title: Monitoring of animal feed contamination by mycotoxins: Results of five years of official control by an accredited Italian laboratory

The manuscript presents well organized description of mycotoxin contamination of feed in an Italian region. Results are based on fully characterized and firmly established analytical methods used for the official control. Discussion is relevant, with justified and logical conclusions, without inconsistencies. The manuscript is well guided and it could be of considerable interest not only for feed scientists but also for feed/food safety authorities, feed producers and farmers¢ associations.

Comments:

Line 57: FUMO – it stands for plural noun?

Line 157: the authors should present the results of the relevant Proficienty testing rounds.

Line 163: along with the number of samples, the authors should provide information related to the time period of sampling. We know that the samples were taken within 5 years perion (title, line 258...) but we do not no when, from-to, which is important for the topic of the manuscript.

Lines 184-185: these lines should be moved in front of Figure 1 (the announcement and the Figure itself are too distant).

Line 186, Table 3: in case of multiple non-compliant results for animal-mycotoxin combination (N° of N.C. for cow-ZEA and swine-DON combination), it would be more logical to such results in ascending order, N° of N.C. for cow-ZEA and swine-DON combinations.

Line 186, Table 3: I would suggest to add a column at the end of Table 3 in order to present total number of samples analyzed and number of non-compliant results per animal specie.

Line 186, Table 3: in the last row of the Table 3, called TOTAL, where authors present total number of samples analyzed on a given mycotoxin, a percentages of contaminated samples and non-compliant samples per given mycotoxin should be added (the authors refer to such percentage in text, in lines 195-195, but not precisely). Percentages of contaminated samples corresponding to animal-mycotoxin combinations are presented on Figure 1, thus they could be omitted from Table 3, as the authors did.

Lines 192-194: please delete these lines, because the same information has been repeated in discussion. Furthermore, the first mentioned number is almost half of the..., but the second is not close but one corresponds to half of the..., as correctly presented in discussion. Additionally, the units for concentration for T-2/HT-2 toxins are micrograms per kg, not miligrams per kg (according to Table 3).

Lines 203, 204, 205, 207, 209: the units for concentration for DON are miligrams per kg, not micrograms per kg (according to Table 3).

Lines 209, 210, 2012: the units for concentration for ZEA are miligrams per kg, not micrograms per kg (according to Table 3).

Figure 2 note: this figure presents data from Table 1 and one sort of data present in Table 3, thus there are some repetitions, but the Figure itself contributes to the visualization of the data and thus enables readers to more efficiently perceive the significance of the results and their relationship to the legal limits.

Line 254-255: Regarding ZEA, the information related to Oceania and North America should be accompanied with one related to Europe (if the cited study provides such info), in order to compare the results of the present study with previous results for the same/close geographical area.

Line 257, Conclusions: could authors include some recommendations based on their results?

Section Results and section Discussion: the authors have not presented results for co-occurrence of mycotoxins, i.e., simultaneous presence of two or more mycotoxins in one sample. Thus, the results could be qualified as underused. The number of samples with multiple mycotoxins, the number and type of mycotoxins in such samples, the most frequent combination of mycotoxins ... should be presented. The non-compliant samples with multiple mycotoxins should also be described in such a way, if there are any.

Author Response

Dear Editors,
We would like to thank you for the timely work carried out in the management of the submitted paper. Please find enclosed the revised version of the manuscript microorganisms-2780545. All the comments and suggestions by Academic Editor and reviewers have been carefully considered and the manuscript has been revised accordingly. Below a detailed point-by-point response to each comment and the relevant changes in the manuscript are listed.
I hope you will find this review suitable for publication in the Microorganisms Special Issueentitled: “Mycotoxins in Food Safety, Food Security and Sustainability".
Sincerely yours,
Dr. Marco Iammarino on behalf of co-authors

The authors would like to thank the Academic Editor and the reviewers for their effort in improving the scientific impact of the Paper. The manuscript has been revised, according to suggestions and comments, editing corrections and rewording the text where necessary. Please note that references to lines are referred to pdf.

Reviewer 1 comments:

The manuscript presents well organized description of mycotoxin contamination of feed in an Italian region. Results are based on fully characterized and firmly established analytical methods used for the official control. Discussion is relevant, with justified and logical conclusions, without inconsistencies. The manuscript is well guided and it could be of considerable interest not only for feed scientists but also for feed/food safety authorities, feed producers and farmers¢ associations.

Line 57: FUMO – it stands for plural noun?

Answer: Thanks for your remark. The sentence has been modified, since “FUMO” stands for “Fumonisins” (line 58).

Line 157: the authors should present the results of the relevant Proficiency testing rounds.

Answer: According to reviewer’s remark, recent outputs of Proficiency test rounds have been listed at lines 183-185.

Line 163: along with the number of samples, the authors should provide information related to the time period of sampling. We know that the samples were taken within 5 years period (title, line 258...) but we do not no when, from-to, which is important for the topic of the manuscript.

Answer: According to reviewer’s suggestion, the time range of samples collection has been specified at line 105.

Lines 184-185: these lines should be moved in front of Figure 1 (the announcement and the Figure itself are too distant).

Answer: Following referee’s suggestion, the reference to Figure 1 has been moved close to the Figure (lines 287-288).

Line 186, Table 3: in case of multiple non-compliant results for animal-mycotoxin combination (N° of N.C. for cow-ZEA and swine-DON combination), it would be more logical to such results in ascending order, N° of N.C. for cow-ZEA and swine-DON combinations.

Answer: Thanks for your indication. The concentrations have been listed in ascending order.

Line 186, Table 3: I would suggest to add a column at the end of Table 3 in order to present total number of samples analyzed and number of non-compliant results per animal specie.

Answer: Thanks for your suggestion. Another column has been added to the right with the indication regarding total number of analysed samples and N.C.

Line 186, Table 3: in the last row of the Table 3, called TOTAL, where authors present total number of samples analyzed on a given mycotoxin, a percentages of contaminated samples and non-compliant samples per given mycotoxin should be added (the authors refer to such percentage in text, in lines 195-195, but not precisely). Percentages of contaminated samples corresponding to animal-mycotoxin combinations are presented on Figure 1, thus they could be omitted from Table 3, as the authors did.

Answer: According to reviewer’s suggestion, the last row was integrated by adding the percentage of samples > LOD, the number of N.C., together with related percentage.

Lines 192-194: please delete these lines, because the same information has been repeated in discussion. Furthermore, the first mentioned number is almost half of the..., but the second is not close but one corresponds to half of the..., as correctly presented in discussion. Additionally, the units for concentration for T-2/HT-2 toxins are micrograms per kg, not miligrams per kg (according to Table 3).

Answer: Following the reviewer’s suggestion the sentence at lines 192-194 has been removed from the text (line 249).

Lines 203, 204, 205, 207, 209: the units for concentration for DON are miligrams per kg, not micrograms per kg (according to Table 3). Lines 209, 210, 2012: the units for concentration for ZEA are miligrams per kg, not micrograms per kg (according to Table 3).

Answer: Thanks for your careful work of revision and please apologize for the oversight. The measurement units have been corrected (lines 259-268).

Figure 2 note: this figure presents data from Table 1 and one sort of data present in Table 3, thus there are some repetitions, but the Figure itself contributes to the visualization of the data and thus enables readers to more efficiently perceive the significance of the results and their relationship to the legal limits.

Line 254-255: Regarding ZEA, the information related to Oceania and North America should be accompanied with one related to Europe (if the cited study provides such info), in order to compare the results of the present study with previous results for the same/close geographical area.

Answer: Thanks for your suggestion. A comment has been added at lines 343-346.

Line 257, Conclusions: could authors include some recommendations based on their results?

Answer: According to referee’s comment, a final recommendation has been added at lines 373-377).

Section Results and section Discussion: the authors have not presented results for co-occurrence of mycotoxins, i.e., simultaneous presence of two or more mycotoxins in one sample. Thus, the results could be qualified as underused. The number of samples with multiple mycotoxins, the number and type of mycotoxins in such samples, the most frequent combination of mycotoxins ... should be presented. The non-compliant samples with multiple mycotoxins should also be described in such a way, if there are any.

Answer: Thanks for your comment. It is worth noting that (as appreciable from table 3) not all mycotoxins were analyzed for all samples. So, the indications about co-occurrence can only be considered as a general remark. A new sentence, specifically focused on the possible co-occurrence of different mycotoxins in the same sample has been added at lines 276-286.

Reviewer 2 Report

Comments and Suggestions for Authors

Notable recommendations

My general impression of the article: the article does not contain any information about the microorganisms themselves at all. It is clear that mycotoxins are synthesized by different mycelial fungi, and they are mentioned by two words in the article (lines 28-32), however the role and presence of the fungi is discussed in many similar articles that present long-term statistics on monitoring mycotoxin concentrations in various animal feeds in different countries. I think this article is more suitable for the Journal Toxins of MDPI. In order for this article to be interesting for readers of Microorganisms Journal, it must contain some interesting information about microorganisms. I recommend that the authors add such information to this article and discuss it.

In general, it is not clear from the article what the composition of the feed was for all these animals? What percentage was corn, what other ingredients were there, what was the moisture content of the feed samples, what was the shelf life and storage conditions. From the microbiological point of view all these facts are important, but they are not in investigation and discussion. Only a comparison with other similar publications in the discussion is not enough to make the article interesting and containing new information. Now the data presented in the article simply confirm statistics what was previously known.

Slight recommendations

The heading of Table 3 indicates that 722 samples of animal feed were examined, and at the end of the table, the Total row indicates the number of samples that contain one or another mycotoxin. Therefore, the percentage of those samples that contain a certain mycotoxin from the total number should be calculated and given in the same table. This will be interesting information.

Lines 206-219: I recommend that the authors remove from the text repetitions of the same mycotoxin concentrations as indicated in Table 3. Here, only the text itself (without specifying concentrations) should be retained with a discussion of how many samples show one or another trend in mycotoxin concentrations compared to legal limits.

The name of Figure 1 should be clarified that these are samples of animal feed, otherwise it looks as if these concentrations of mycotoxins were found not in the feed, but in the organisms of the animals themselves.

Author Response

Dear Editors,

We would like to thank you for the timely work carried out in the management of the submitted paper. Please find enclosed the revised version of the manuscript microorganisms-2780545. All the comments and suggestions by Academic Editor and reviewers have been carefully considered and the manuscript has been revised accordingly. Below a detailed point-by-point response to each comment and the relevant changes in the manuscript are listed.

I hope you will find this review suitable for publication in the Microorganisms Special Issue entitled: “Mycotoxins in Food Safety, Food Security and Sustainability".

Sincerely yours,

Dr. Marco Iammarino on behalf of co-authors

The authors would like to thank the Academic Editor and the reviewers for their effort in improving the scientific impact of the Paper. The manuscript has been revised, according to suggestions and comments, editing corrections and rewording the text where necessary. Please note that references to lines are referred to pdf.

Reviewer 2 comments:

My general impression of the article: the article does not contain any information about the microorganisms themselves at all. It is clear that mycotoxins are synthesized by different mycelial fungi, and they are mentioned by two words in the article (lines 28-32), however the role and presence of the fungi is discussed in many similar articles that present long-term statistics on monitoring mycotoxin concentrations in various animal feeds in different countries. I think this article is more suitable for the Journal Toxins of MDPI. In order for this article to be interesting for readers of Microorganisms Journal, it must contain some interesting information about microorganisms. I recommend that the authors add such information to this article and discuss it.

In general, it is not clear from the article what the composition of the feed was for all these animals? What percentage was corn, what other ingredients were there, what was the moisture content of the feed samples, what was the shelf life and storage conditions. From the microbiological point of view all these facts are important, but they are not in investigation and discussion. Only a comparison with other similar publications in the discussion is not enough to make the article interesting and containing new information. Now the data presented in the article simply confirm statistics what was previously known.

Answer: Thanks for your comment. We agree with the referee. For this reason, the manuscript was submitted to the special issue entitled: “Mycotoxins in Food Safety, Food Security and Sustainability” which is devoted to this topic.

Following the reviewer’s comment, more info and literature regarding microorganisms has been added in the Introduction (lines 48-69).

Regarding feed samples composition, taking into account the large number of samples analyzed, this information is very variable, so that the addition of clear info is very difficult. Thus, the authors opted to add most significant information related to feed composition only for non-compliant samples (lines 258-274)

Slight recommendations

The heading of Table 3 indicates that 722 samples of animal feed were examined, and at the end of the table, the Total row indicates the number of samples that contain one or another mycotoxin. Therefore, the percentage of those samples that contain a certain mycotoxin from the total number should be calculated and given in the same table. This will be interesting information.

Answer: Thanks for your suggestion. Another row has been added at the end of table 3, indicating the percentage of samples > LOD, the number of N.C., together with related percentage.

Lines 206-219: I recommend that the authors remove from the text repetitions of the same mycotoxin concentrations as indicated in Table 3. Here, only the text itself (without specifying concentrations) should be retained with a discussion of how many samples show one or another trend in mycotoxin concentrations compared to legal limits.

Answer: Thanks for your comment. A sentence has been removed (line 249), since the same comment is already reported in discussion. Regarding the rest of this section, the authors would prefer to maintain it, since significant info regarding samples composition is specified.

The name of Figure 1 should be clarified that these are samples of animal feed, otherwise it looks as if these concentrations of mycotoxins were found not in the feed, but in the organisms of the animals themselves.

Answer: Thanks for your remark. Figure caption modified according to suggestion. (line 304).

Round 2

Reviewer 2 Report

Comments and Suggestions for Authors

Thanks to the authors for taking into account my previous comments and doing the necessary work with the text.

Now I recommend the authors:

- proofread the text again and remove the remaining typos in it (they were added with the new introduced text);

- add in Figure 2 the limits of variation of all the indicated average values, since now they are point values, but in fact they represent the average values calculated by the authors, and it should be indicated within what limits they vary. Table 3 shows the minimum and maximum values, but this is not the same as the deviations from the average value obtained. Please make the necessary addition.

Comments on the Quality of English Language

Please, see my comment for the authers.

Author Response

Dear Editors,

below a detailed point-by-point response to minor revisions requested by referee n.2.

I hope you will find this article suitable for publication in the Microorganisms Special Issue entitled: “Mycotoxins in Food Safety, Food Security and Sustainability".

Sincerely yours,

Dr. Marco Iammarino on behalf of co-authors

Reviewer 2 comments:

Thanks to the authors for taking into account my previous comments and doing the necessary work with the text.

Now I recommend the authors:

- proofread the text again and remove the remaining typos in it (they were added with the new introduced text);

Answer: According to the reviewer’s comment, the document has been spell-checked and some typos have been corrected. Table 3 has been reviewed as well, and added in the main text.

- add in Figure 2 the limits of variation of all the indicated average values, since now they are point values, but in fact they represent the average values calculated by the authors, and it should be indicated within what limits they vary. Table 3 shows the minimum and maximum values, but this is not the same as the deviations from the average value obtained. Please make the necessary addition.

Answer: Thanks for your comment. We apologize for the oversight. Indeed, the revised Figure 2 has been attached as supplementary file (Figures at high resolution) and not in the main text. The revised Figure 2 has now been included in the main text.
